# The Triad of Blood–Brain Barrier Integrity: Endothelial Cells, Astrocytes, and Pericytes in Perinatal Stroke Pathophysiology

**DOI:** 10.3390/ijms26051886

**Published:** 2025-02-22

**Authors:** Tania Garcia-Martínez, Denise G. Gornatti, Marina Ortiz, Guillem Cañellas, Damià Heine-Suñer, Cristòfol Vives-Bauzà

**Affiliations:** 1Neurobiology, Research Unit, Hospital Universitari Son Espases, Health Research Institute of Balearic Islands (IdISBa), 07120 Palma, Spain; tania.garcia@idisba.es (T.G.-M.); denisegala.gornatti@idisba.es (D.G.G.); marina.ortiz@idisba.es (M.O.); guillem.canellas@idisba.es (G.C.); 2Department of Biology, University of Balearic Islands (UIB), Institut Universitari d’Investigacions en Ciències de la Salut (IUNICS), 07122 Palma, Spain; 3Genomics of Health Research, Unit of Molecular Diagnostics and Clinical Genetics, Hospital Universitari Son Espases, Health Research Institute of Balearic Islands (IdISBa), 07013 Palma, Spain; damian.heine@ssib.es

**Keywords:** fetal, pediatric, perinatal stroke, neurovascular unit, blood–brain barrier, vascular brain development

## Abstract

Pediatric stroke, a significant cause of long-term neurological deficits in children, often arises from disruptions within neurovascular unit (NVU) components. The NVU, a dynamic ensemble of astrocytes, endothelial cells, pericytes, and microglia, is vital for maintaining cerebral homeostasis and regulating vascular brain development. Its structural integrity, particularly at the blood–brain barrier (BBB), depends on intercellular junctions and the basement membrane, which together restrict paracellular transport and shield the brain from systemic insults. Dysfunction in this intricate system is increasingly linked to pediatric stroke and related cerebrovascular conditions. Mutations disrupting endothelial cell adhesion or pericyte–endothelial interactions can compromise BBB stability, leading to pathological outcomes such as intraventricular hemorrhage in the germinal matrix, a hallmark of vascular brain immaturity. Additionally, inflammation, ferroptosis, necroptosis, and autophagy are key cellular processes influencing brain damage and repair. Excessive activation of these mechanisms can exacerbate NVU injury, whereas targeted therapeutic modulation offers potential pathways to mitigate damage and support recovery. This review explores the cellular and molecular mechanisms underlying NVU dysfunction, BBB disruption, and subsequent brain injury in pediatric stroke. Understanding the interplay between genetic mutations, environmental stressors, and NVU dynamics provides new insights into stroke pathogenesis. The susceptibility of the germinal matrix to vascular rupture further emphasizes the critical role of NVU integrity in early brain development. Targeting inflammatory pathways and cell death mechanisms presents promising strategies to preserve NVU function and improve outcomes for affected neonates.

## 1. Introduction

Brain development involves complex processes such as neuronal and glial growth, myelination, and the formation of cortical structures, which are guided by conserved genetic mechanisms across mammals. While animal studies have advanced understanding of these processes, humans exhibit unique, species-specific brain features that support higher-order functions [1]. Disruptions in these genetic programs, especially in preterm infants, can lead to neurodevelopmental and psychiatric disorders, with the brain’s maturity at birth being a critical factor [2].

Preterm birth, defined as occurring before 37 weeks of gestation, varies by gestational age: moderate or late preterm (32–37 weeks), very preterm (28–32 weeks), and extremely preterm (less than 28 weeks) [3]. Globally, about 15 million preterm births occur annually, representing 10% of live births. The incidence and causes differ significantly across racial, ethnic, and geographic groups [4].

The NVU and germinal matrix (GM) are essential for neurogenesis, vascular development, and brain maturation during the fetal and neonatal periods, making them highly susceptible to injury during perinatal stroke. The dynamic nature of brain development influences the severity and progression of damage after an ischemic or hemorrhagic event. The stage of postnatal brain maturation and the development pace of central nervous system (CNS) cell types at the time of stroke result in distinct excitotoxic and inflammatory responses. Therefore, it is crucial to target pathophysiological pathways in a maturation-specific manner to protect brain function and support repair in the developing brain after stroke [5].

## 2. Pediatric Stroke

Pediatric stroke is a serious neurological condition that can result in long-term morbidity, including sensorimotor deficits, intellectual disability, and epilepsy [6]. The risk of pediatric stroke is highest during the fetal or neonatal period and is influenced by neuroplasticity, with recovery patterns shaped by the timing, location, and associated comorbidities of the stroke [7].

Pediatric strokes are categorized by age into perinatal strokes (20 weeks gestation to 28 days after birth) and childhood strokes (28 days to 18 years) [8,9,10]. Perinatal strokes include acute perinatal stroke (APS), evident shortly after birth, and presumed perinatal stroke (PPS), diagnosed later due to symptoms like seizures or early handedness [8]. Unlike older patients, perinatal strokes lack clear clinical signs and require neuroimaging for diagnosis.

Pediatric strokes, like those in adults, are classified by cause as either ischemic or hemorrhagic [11]. Ischemic strokes include arterial ischemic stroke (AIS) and venous infarction, typically caused by cerebral or cortical vein thrombosis (CSVT) [8,9]. These strokes are further categorized by timing: fetal ischemic stroke (diagnosed before birth), neonatal ischemic stroke (within 28 days of birth), and presumed perinatal ischemic stroke (PIS, diagnosed later but presumed to have occurred between the 20th week of gestation and 28 days postnatal) [12,13]. Hemorrhagic strokes, on the other hand, can manifest as intracerebral (ICH), intraventricular (IVH), or subarachnoid (SAH) hemorrhage. They are classified as either primary (resulting from vascular anomalies or bleeding disorders) or secondary (arising from ischemic infarction) [8,14]. Distinguishing these stroke types may be challenging without prompt imaging, particularly in neonates, where acute presentations may be missed [12,15,16]. Recognizing stroke at different ages is crucial for timely intervention, as early treatment can help preserve brain function and support recovery. The impact of any stroke is influenced by the neurodevelopmental stage at the time of the event [7], underscoring the importance of early detection and management by pediatric health professionals.

The annual incidence of pediatric stroke ranges from 3 to 25 per 100,000 children in developed countries [8], with a higher rate in neonates (25–40 per 100,000) and even greater in premature neonates (approximately 100 per 100,000) [9]. Perinatal stroke is a leading cause of hemiparetic cerebral palsy and has lifelong impacts, including motor, cognitive, and epileptic issues [10]. Stroke is more common in boys than girls and in Black, Asian, and Hispanic children compared to White children [17,18].

Risk factors for perinatal stroke include maternal conditions (e.g., preeclampsia, infections, diabetes) and neonatal factors (e.g., cardiac disease, prematurity, clotting disorders), though a direct cause is rarely identified [19]. For childhood stroke, common causes are arteriopathy, congenital heart disease, and prothrombotic disorders, with additional risks like infection, trauma, and sickle cell disease. Pediatric stroke differs from adult stroke not only in demographics and risk factors but also in the unique vulnerability and recovery patterns of the developing brain [19,20].

## 3. Human Brain Development

Comprehensive reviews detailing early neurogenesis in the perinatal brain have been recently published [1,21,22]. Human brain development begins around the second to third gestational week (GW) and continues through childhood, orchestrated by a complex interplay of genetic, epigenetic, and environmental factors. Early in the first trimester, the process of neurulation sets the stage for CNS formation as neuroepithelial cells in the ectoderm form the neural plate. By the third GW, the plate folds and fuses into a closed neural tube, marking the foundation of the CNS [23,24]. Shortly after, the brain’s vasculature develops, separating the CNS from the periphery around the fifth GW [21] (Figure 1).

As the neocortex begins to form between the eighth and eighteenth GW, neurogenesis intensifies. Neural epithelial cells (NECs) initially divide symmetrically to expand their population, then switch to asymmetric divisions to produce the first neurons. These early neurons migrate radially into a structure called the preplate [25,26] (Figure 1a). Over time, NECs evolve into radial glia cells, which act as scaffolds for the migration of excitatory neurons to the developing cortical layers, while inhibitory neurons migrate tangentially into the cortex. By the ninth to thirteenth GWs, neurogenesis peaks, supported by intermediate progenitor cells in the subventricular zone [27] (Figure 1a). In the third trimester, neuronal axons and glial cells, such as astrocytes and oligodendrocytes, are produced and integrated into neural circuits [28]. Synaptogenesis, the formation of synaptic connections, begins as early as the eighth GW [29], with microglia and astrocytes shaping neural networks through synaptic pruning [30,31]. Microglia, derived from the hematopoietic lineage, appear by the fifth GW (Figure 1b) and contribute to early brain development even before vascular sprouting [32]. Astrocytes and oligodendrocyte progenitors emerge later, between the fifteenth and seventeenth GW [33,34]. Myelination, the process of insulating nerve fibers, starts mid-gestation and continues into late adolescence, supporting the brain’s growing complexity and efficiency (Figure 1b). Postnatally, the brain’s structure and plasticity continue to evolve, enabling lifelong adaptation and learning [2,5].

## 4. Human Vascular Development

The development and expansion of vascular beds are essential for embryonic growth, as vascular defects during this stage can result in severe brain malformations [35]. In humans, brain vascularization progresses in parallel with brain development, following a precise sequence of angiogenic processes. By GW eight, the primitive capillary network in the perineural vascular plexus begins to expand into the CNS parenchyma [36] (Figure 1).

During early neuroepithelial development, endothelial cells (ECs) originate from mesodermal angioblasts and migrate into the neuroepithelium via the perineural vascular plexus, guided by Vascular Endothelial Growth Factor (VEGF) [37]. Alongside VEGF, several key signaling pathways contribute to angiogenic sprouting and vessel remodeling, including the Angiopoietin–Tie system, the Hedgehog pathway, Ephrin receptor tyrosine kinases, Dll4/Notch1 signaling, and the Wnt signaling pathway [38,39,40,41,42,43]. At this stage, the developing blood vessels exhibit short, primitive tight junctions (TJs), lack fenestrations, and limit the movement of blood-derived proteins into the CNS environment [37]. TJ proteins, such as occludin and claudin-5, critical for CNS barriers, begin to emerge by the sixteenth GW [5,36]. As development progresses, ECs aggregate into primitive structures that mature into functional blood vessels [44].

The formation of vascular networks occurs through sprouting angiogenesis, where new blood vessels grow outward and connect to other vessels or sprouts via anastomosis. At the cellular level, this process is characterized by dynamic and highly regulated behaviors (reviewed by [45,46]). In response to pro-angiogenic signals, ECs are activated, remodeling their actin cytoskeletons and downregulating junctional adhesion molecules to enable cell motility. Throughout this process, ECs maintain contact with neighboring cells and the parent vessel, ensuring vascular integrity and proper apical-basal cell polarization [47]. Fine-tuning these cellular responses to angiogenic signals is crucial for the successful formation and maturation of vascular networks.

## 5. Neurovascular Unit Cell–Cell Interaction

Two comprehensive reviews detailing the cellular components and function of the NVU in the perinatal brain were recently published [48,49]. The NVU is formed by interactions between blood vessels and neural and glial cells in the CNS, including microvascular ECs, neurons, microglia, astrocytes, pericytes, and matrisomes [50,51] (Figure 2). These components engage in close and intricate interactions, allowing the NVU to function as a cohesive unit. The NVU maintains a highly selective BBB, ensuring cerebral homeostasis, regulates local blood flow through neurovascular coupling, provides trophic support, produces growth factors and paracrine signals, guides neuronal differentiation, and recruits oligodendrocytes [48,52,53,54]. 

The NVU plays a crucial role in regulating the local blood supply to meet neuronal metabolic needs by adjusting the diameter of blood vessels [48,55]. In this process, astrocytes and pericytes are activated by glutamate released from neurons. Glutamate, in turn, triggers the release of vasoactive mediators that modulate vascular tone (Figure 2b). The balance between vasoconstrictors and vasodilators controls the tone of the surrounding vasculature, thereby regulating local blood flow [48].

The second major function of the NVU is maintaining the BBB [48]. The BBB serves as both a physical and metabolic barrier between the bloodstream and neuroglia [55]. It acts as a physical barrier due to the presence of complex TJs [49,51] and adherent junctions (AJs) between ECs [55]. This forces most molecular traffic to traverse the BBB via a transcellular route rather than passing between cells via the junctions typically found in other endothelial tissues [51]. Consequently, paracellular transport of water-soluble or polar compounds is limited, small ion transport is restricted, and EC polarity is preserved [55]. The role of the BBB depends on the interactions and coordination among various cellular components and the extracellular matrix (ECM), which together create a functional and anatomical barrier system [56].

## 6. Astrocyte, Pericytes, and ECs Interactions

The BBB is established and maintained through intricate interactions between ECs, pericytes, and astrocytes. These cellular components of the NVU work together to regulate angiogenesis and ensure the barrier’s protective and selective permeability, which is critical for brain function [57] (Figure 2).

During late embryogenesis and postnatal development, the BBB undergoes progressive maturation. TJs formed by ECs become increasingly complex, creating impermeable barriers essential for maintaining brain homeostasis. By GW12 in humans, proteins such as occludin, claudins, and junctional adhesion molecules (JAMs) organize into functional TJs, linking to the cytoskeleton via zonula occludens (ZO)-1, ZO-2, and ZO-3 [58]. AJs, composed of VE-cadherin, PECAM-1, and nectin, provide structural stability, while gap junctions (GJs), formed by connexin hemichannels, facilitate direct intercellular communication [59] (Figure 2b). These features mature by embryonic day 15 (E15) in rodents, and by E19, TJs are fully differentiated [60], setting the stage for robust BBB functionality [61,62]. Astrocytic contributions to TJ tightness start closer to birth and become more prominent during postnatal development [63].

Both ECs and pericytes are situated in the basement membrane (BM) and anchored to it via integrins (Figure 2b). Pericytes play essential roles in BBB development and maintenance. They promote TJ formation [61], regulate vascular development [64], and contribute to inflammatory processes [65]. Signaling between ECs and pericytes involves platelet-derived growth factor BB (PDGF-BB) and its receptor (PDGFR-β), adhesion via N-cadherin, and GJs that facilitate the exchange of ions and small molecules [66,67] (Figure 2b). These interactions ensure BBB integrity and permeability control during embryogenesis and adulthood. While pericytes are indispensable, studies suggest that CNS-derived signals can independently induce BBB-specific molecule expression in ECs, highlighting additional regulatory mechanisms [43,61].

Astrocytes, located furthest from the vascular lumen within the NVU, interface with the BM and ECM through their endfeet, contributing ECM proteins and regulating vascular tone via calcium signaling [57] (Figure 2). They influence TJ protein expression by secreting factors like Src-suppressed C-kinase substrate (SSeCKS) and maintain the integrity of the BBB by modulating matrix metalloproteinase-9 (MMP-9) activity in pericytes [68,69]. Apolipoprotein E (ApoE3), secreted by astrocytes, binds to low-density lipoprotein receptor-related protein 1 (LRP1) in pericytes, suppressing the pro-inflammatory CypA-NF-κB-MMP-9 pathway [69]. This prevents the degradation of TJ and BM proteins, protecting the BBB, whereas ApoE4 lacks this protective effect [69]. Astrocytes also metabolize glucose taken up by ECs via Glut1, converting it to lactate, which is then supplied to neurons through monocarboxylate transporters MCT1 and MCT4 [59] (Figure 2b).

ECs, integral to the BBB, form the physical barrier and facilitate selective nutrient exchange. Their connections through TJs, AJs, and GJs are critical for maintaining the barrier’s integrity. Knockout studies reveal that astrocyte-derived laminin is essential for TJ protein expression and overall BBB functionality [70]; its absence leads to barrier leakage [71], particularly under pathological conditions such as perinatal stroke.

## 7. Mechanisms of Brain Damage After a Perinatal Stroke

### 7.1. Pathophysiological Mechanisms of a Hypoxic–Ischemic Injury

Hypoxic–ischemic (HI) injury during the perinatal period triggers a cascade of cellular and molecular processes, leading to necrotic and apoptotic cell death. Unlike in adults, the BBB in neonates is not fully developed, contributing to the unique pathophysiology of neonatal strokes. Despite its early developmental stage, studies show that the BBB remains largely impermeable to very small molecules, maintaining a degree of selective permeability [72,73].

HI injury results in neuronal death through two primary phases [74] (Figure 3): First, a primary energy failure occurs due to the deprivation of oxygen and glucose, which disrupts normal metabolic processes. This disruption leads to anaerobic metabolism, intracellular lactate accumulation, and the production of reactive oxygen species (ROS), ultimately resulting in necrotic cell death. For several cells essential for maintaining CNS homeostasis—such as pericytes, astrocytes, microglia, and ECs—damage to the BBB plays a critical role in the pathogenesis of NVU dysfunction, subsequently causing downstream perivascular damage [70,75]. Furthermore, the uncontrolled release of excitatory neurotransmitters exacerbates neuronal damage [73,75]. Second, if the insult persists, it leads to secondary energy failure, which occurs at least 6 h post-injury and may continue for days after the initial event. During this phase, most cell deaths result from apoptosis, driven by excitotoxicity, oxidative stress, mitochondrial dysfunction, and inflammation [73,74,76].

After the acute injury phase, the chronic phase is characterized by astrogliosis and chronic inflammation, with processes aimed at restoring CNS integrity, although often incomplete or maladaptive [9,70,74] (Figure 3).

Neonatal stroke triggers distinct immune responses compared to adults due to the immaturity of immune pathways and age-dependent mediator effects. Microglia play a dual role in neonatal HI injury. Initially, they exacerbate damage, but later, they contribute to recovery through processes such as debris clearance and extracellular matrix remodeling. Following injury, microglia are activated via toll-like receptors (TLRs) in response to damage-associated molecular patterns (DAMPs), transitioning from a resting to an active state [77] (Figure 4a). These cells polarize into either pro-inflammatory (M1) or anti-inflammatory (M2) phenotypes. M1 microglia release damaging inflammatory mediators, while M2 microglia promote repair through cytokines and neurotrophic factors [77]. This dynamic polarization shifts from M1 dominance during the acute phase of injury to M2 activity during recovery. Imbalances in this transition impact neonatal HI injury outcomes (Figure 4a).

Like microglia, astrocytes also exhibit two polarization states in response to HI injury: neurotoxic A1 (pro-inflammatory) and neuroprotective A2 (anti-inflammatory). Pro-inflammatory astrocytes contribute to BBB disruption by increasing the expression of factors like VEGF, cytokines (e.g., TNF-α, IL-1β), ROS, and MMPs. Additionally, astrocytes recruit immune cells by up-regulating ICAM-1 and VCAM-1 expression and activating microglia, thereby indirectly intensifying inflammation-induced BBB damage (reviewed in [78]). Following stroke, pro-inflammatory astrocytes, together with infiltrating leukocytes and activated microglia, secrete matrix metalloproteinases (MMPs) [79,80]. Particularly, MMP-9, MMP-3, and MMP-2 mediate ECM breakdown, degrading TJs, laminin, collagen, and fibronectin. This degradation results in vasogenic edema, BBB leakage, and immune cell infiltration [81,82,83,84,85] (Figure 4). Upregulation of MMP-9 has been observed in neonatal rodent brains 24 h after ischemic injury [86]. Moreover, MMPs facilitate adhesion molecule signaling and ECs activation, processes critical for monocyte recruitment and the inflammatory cascades seen in neonatal brain injuries [5,87,88]. Anti-inflammatory astrocytes, in contrast, promote BBB repair by resolving inflammation. They secrete protective cytokines like IL-2, IL-10, and TGF-β and release PTX3 and IGF-1, which protect the BBB and support immune cell polarization toward anti-inflammatory states (reviewed in [78]) (Figure 4a). Anti-inflammatory astrocytes also release MMPs, but those instead promote tissue repair [89,90]. Moreover, anti-inflammatory astrocytes also regulate microglial activation via CX3CR1 and IL-4Rα signaling [91] (Figure 4a). Notably, neuroprotective astrocytes, which dominate three days post-stroke, support ECM integrity and scar formation through astrogliosis. This process limits immune cell migration and confines inflammation within the infarct region [92].

Pericytes, another key cellular responder to ischemic stroke damage, also exhibit remarkable plasticity under ischemic conditions. They can transition into microglia-like or other cell types through a multipotent stem cell intermediate state, meaning their loss worsens the leukocyte infiltration [93,94,95].

### 7.2. Pathophysiological Mechanisms of Brain Damage After Intracerebral Hemorrhage (ICH)

ICH triggers complex pathological processes, resulting in acute and delayed brain damage. These processes include mechanical injury, inflammatory and immune responses, BBB disruption, oxidative stress, and cell death, with significant implications for perihematomal edema and long-term functional outcomes [96,97] (Figure 3). ICH initially causes mechanical injury through increased intracranial pressure and reduced local cerebral blood flow, leading to ischemia and brain tissue compression. This mechanical stress exacerbates secondary damage by promoting edema formation and cytotoxicity [96]. The activation of the coagulation cascade is a major contributor to early edema formation after ICH. Thrombin, generated by the breakdown of prothrombin due to BBB disruption, induces BBB dysfunction by activating Src kinase phosphorylation, damaging brain microvascular ECs, and astrocytes [97]. While Src kinase activity initially exacerbates injury, it later contributes to BBB restoration and edema resolution after 2–6 days [98]. Thrombin also activates the complement cascade, leading to the release of nitric oxide (NO), tumor necrosis factor-alpha (TNF-α), and interleukins (e.g., IL-6, IL-12), which amplify inflammatory responses and promote secondary injury [98] (Figure 4b).

Around three days post-ICH, delayed edema arises, primarily mediated by red blood cell breakdown and hemoglobin degradation products, such as heme and iron. These compounds, along with enzymes like carbonic anhydrase, contribute to oxidative stress and further edema development [97,99] (Figure 3 and Figure 4b).

The inflammatory response is central to ICH pathophysiology, involving both cellular (microglia, astrocytes, macrophages, leukocytes) and molecular (cytokines, chemokines, reactive oxygen species, extracellular proteases) mediators [99,100]. Pro-inflammatory cytokines, such as IL-1β, IL-17, and TNF-α, exacerbate injury by amplifying inflammation and increasing BBB permeability (Figure 4b). Conversely, anti-inflammatory cytokines (e.g., IL-4, IL-10, TGF-β) facilitate tissue repair [100]. IL-6, a dual-function cytokine, participates in both pro- and anti-inflammatory pathways depending on the stage of injury [100] (Figure 4b). Neuroinflammation after IHC also involves a complex interplay of M1 and M2 macrophages/microglia. While M1 cells drive detrimental neuroinflammation, M2 cells promote hematoma resolution, edema resorption, and white matter repair [101]. The NLRP3 inflammasome, a key protein complex in inflammation following ICH, is activated in M1 microglia through mechanisms such as the complement system. Once activated, it triggers caspase-1, which converts pro-IL-1β and pro-IL-18 into their active forms, IL-1β and IL-18, leading to pyroptosis and elevated pro-inflammatory cytokines [100]. Neutrophils also exert dual roles, contributing to ICH growth and injury but aiding recovery through lactoferrin-mediated iron scavenging and edema reduction [100] (Figure 4b).

As mentioned, the BBB undergoes significant dysfunction in ICH, exacerbating edema formation and allowing for infiltration of immune cells and neurotoxic molecules. Treatments like hyperosmotic solutions (e.g., mannitol) and glibenclamide, a sulfonylurea receptor inhibitor, show promise in mitigating perihematomal edema [102].

Cell death pathways associated with ICH include autophagy, ferroptosis, and necroptosis. Oxidative stress, driven by hemoglobin-derived iron and lipid peroxidation, plays a central role in neuronal and vascular injury. Ferroptosis, a non-apoptotic, iron-dependent form of cell death mediated by glutathione peroxidase 4 (GPX4), is a key contributor [103]. Experimental studies demonstrate that lipid peroxidation inhibitors, such as ferrostatin-1 and liproxstatin-1, reduce ICH-induced brain atrophy, neuronal death, and functional deficits [103].

## 8. Emerging Cell-Based Therapeutic Strategies for Pediatric Stroke

Strategies aimed at promoting the anti-inflammatory polarization of both microglia and astrocytes or inhibiting harmful pro-inflammatory responses hold promise for the management of both ischemic and hemorrhagic pediatric stroke. In line with these strategies, cell-based treatments using mesenchymal stem cells (MSCs) or umbilical cord blood-derived mesenchymal stem cell (UC-MSCs) transplantation show potential, as they possess immunosuppressive and anti-inflammatory properties that may have protective and regenerative effects (reviewed in [104,105]). MSCs modulate the immune response by regulating the function of immune cells, such as T cells and B cells, macrophages, and dendritic cells [106]. In neonatal rodents, intraventricular transplantation of UC-MSCs has been shown to reduce post-hemorrhagic hydrocephalus and brain injury following IVH [107]. In addition to their anti-inflammatory properties, studies in animal models have demonstrated that both MSCs and UC-MSCs promote neuroplasticity by activating NMDA and AMPA receptors. This activation, in turn, stimulates the expression of various trophic factors, such as VEGF [108,109]. Through this mechanism, MSCs and UC-MSCs may also contribute to angiogenesis, EC proliferation, and astrocyte differentiation processes known to be regulated by VEGF. Furthermore, MSCs also exert neuroprotective effects through mitochondrial transfer and the regulation of ROS production [110].

Other cells with neuroprotective potential include endothelial progenitor cells (EPCs), which secrete protective cytokines and growth factors. These factors support the self-repair of injured ECs and help restore their structure and function by integrating them into damaged areas [111]. Another promising cell source for therapy is induced pluripotent stem cell (iPSC)-derived progenitors. The administration of iPSC-derived neural progenitor and precursor cells has been shown to enhance sensorimotor function, reduce lesion volume, and promote neurogenesis and angiogenesis [112]. Additionally, positive effects have been observed with iPSC-derived MSCs, inducing angiogenesis [113], as well as with neuroepithelial-like stem cells, which promote neuronal regeneration [114]. Overall, regenerative cells may contribute to a favorable environment for tissue repair, ultimately improving functional outcomes after perinatal stroke.

## 9. NVU and Monogenic Neurological Disorders

Several genetic diseases originate within specific cell types of the NVU and are closely linked to critical roles in BBB development, function, and regulation. Although these diseases are rare, they provide valuable insights into pathogenic mechanisms and causal relationships. Below, we outline some of these disorders, the associated cell types, and the potential pathogenic role of BBB dysfunction.

### 9.1. Pediatric Stroke Caused by Mutations in Genes Expressed in ECs

Mutations in genes critical for ECs and the endothelium-derived ECM play a significant role in brain hemorrhage. These genes encode proteins vital for the structure and regulation of EC-cell junctions, vascular basement membranes, and transporters essential for BBB integrity. Biallelic variants in TJ-associated genes, such as JAM2 [115], JAM3 [72], and OCLN (occludin) [116,117,118,119], have been linked to increased BBB permeability, resulting in uncontrolled leakage of solutes and plasma proteins, along with excessive trans-endothelial leukocyte migration. This disruption leads to neuroinflammation. JAM-C normally inhibits this process [120]. Patients harboring such mutations often present with brain hemorrhage, calcification, movement disorders, and cognitive or neurobehavioral abnormalities [121,122].

The ECM and basement membranes not only provide structural support but also act as reservoirs for growth factors that regulate cellular functions. Type IV collagens, a major component of basement membranes, are vital for preserving BBB structure and function. For instance, mutations in the α1(IV) chain (*COL4A1*) cause perinatal cerebral hemorrhage [123].

Mutations in *COL4A1/A2* manifest as lacunar ischemic strokes, deep ICH, and white matter hyperintensities [59]. A murine model demonstrated that *Col4A1* mutations predisposed both neonatal and adult mice to ICH. Durrani et al. (2017) reported a term infant with extensive intrauterine stroke, encephalomalacia, and anterior segment dysgenesis caused by a de novo mutation in *COL4A1* [124].

Intracranial or intraventricular hemorrhage (ICH/IVH) is a significant type of perinatal stroke, particularly in preterm infants. These hemorrhages originate in the GM, a highly vascularized region located between the caudate nucleus and the thalamus near the foramen of Monro. The GM microvasculature is especially fragile due to immature basal lamina, sparse pericyte coverage, deficient TJs, and limited astrocytic support [125] (Figure 2a). Lecca et al. identified bi-allelic loss-of-function mutations in the *ESAM* gene in a cohort of thirteen perinatal stroke cases, including four fetuses, from eight unrelated families [122]. Furthermore, four additional perinatal stroke patients from two unrelated families have recently been reported to harbor homozygous *ESAM* variants, including the first described missense mutation [126]. ESAM (endothelial cell adhesion molecule), a TJ protein critical for BBB integrity, was not previously linked to rare human disease traits. Patients harboring *ESAM* mutations exhibit severe global developmental delay, epilepsy, spasticity, ventriculomegaly, ICH, and cerebral calcifications. This phenotype closely resembles conditions associated with other TJ-related gene mutations, such as JAM2, JAM3, and OCLN [117,118,119,127]. ESAM, a member of the immunoglobulin receptor family, facilitates endothelial cell adhesion through homophilic interactions. Its expression is predominantly restricted to embryonic and adult vasculature, where it regulates endothelial permeability and neutrophil extravasation. Research by Sauteur et al. (2017) highlighted the overlapping but distinct roles of ESAM and VE-cadherin in blood vessel formation and endothelial cell recognition during vascular anastomosis [47]. In a murine model, Duong et al. (2020) demonstrated that ESAM is crucial for maintaining endothelial junction integrity, particularly in the lungs. Gene inactivation of *Esam* increased vascular permeability in the lung, though it did not affect the heart, skin, or brain [128]. 

Additionally, vascular malformations, including cerebral cavernous malformations (CCMs) and arteriovenous malformations (AVMs), significantly increase ICH risk. CCMs are characterized by fragile venous capillary clusters lacking smooth muscle support, rendering them prone to rupture. Genetic mutations in *KRIT1*/*CCM1*, *CCM2*, and *CCM3* underlie most CCMs [129]. AVMs, congenital anomalies of direct arterial-to-venous connections, are driven by excessive VEGF signaling and mutations in RAS/MAPK pathway genes like *KRAS*. Hereditary hemorrhagic telangiectasia (HHT) is a classic example of a syndrome associated with AVMs. Most individuals meeting the clinical diagnostic criteria for HHT harbor pathogenic variants in four TGFb pathway genes: Endoglin (*ENG)*, Activin Receptor-Like 1 (*ACVRL1* or *ALK1*), SMAD family member 4 (*SMAD4*), and Growth Differentiation Factor 2 (*GDF2* or *BMP9*)] [130]. Another genetic disorder involving AVMs is capillary malformation–arteriovenous malformation (CM-AVM) syndrome, which follows an autosomal dominant inheritance pattern due to loss of function variants in RAS P21 Protein Activator 1 (*RASA1)* and Ephrin Receptor B4 (*EPHB4*)] [131,132].

Sturge Weber Syndrome (SWS), a rare neurocutaneous disorder, also contributes to ICH risk through vascular malformations. Unlike inherited disorders such as HHT and CM-AVM, SWS is caused by a somatic activating mutation in *GNAQ*, which encodes the G protein subunit alpha-q (Gαq). This mutation, *GNAQ* p.R183Q, located in the switch 1 domain of Gαq, leads to constitutive activation of the protein and is present in approximately 88% of brain lesions in SWS patients [133,134]. SWS is characterized by abnormal capillary and venous blood vessels, with associated leptomeningeal vascular malformations.

This array of vascular malformations highlights the diverse genetic underpinnings contributing to ICH risk, emphasizing the importance of mutations in genes regulating endothelial cell function, vascular development, and inflammatory processes.

### 9.2. Pediatric Stroke Caused by Mutations in Genes Expressed in Vascular Mural Cells and Pericytes

Cerebral autosomal dominant arteriopathy with subcortical infarcts and leukoencephalopathy (CADASIL) is a relatively common autosomal-dominant stroke syndrome, affecting approximately 2–4 per 100,000 individuals. It is caused by mutations in the *NOTCH3* gene, which is specifically expressed in vascular mural cells. These cells include vascular smooth muscle cells (VSMCs) and pericytes, playing a vital role in maintaining the integrity of blood vessels. Over 200 Notch3 mutations have been associated with CADASIL, though the causative role of loss-of-function mutations remains uncertain [135]. The CADASIL phenotype is predominantly associated with the expression of Notch3 in VSMCs, suggesting a key role in vascular health. Studies in *Notch3^−/−^* mice have demonstrated a progressive decline in VSMCs, mirroring the vascular pathology observed in CADASIL [136]. This regression of VSMCs is associated with decreased vessel wall thickness, loss of extracellular matrix, and weakening of the vessel wall. A key hypothesis posits that Notch3 facilitates communication and cell–cell interactions between VSMCs and arterial ECs, which is crucial for vascular stability. Experimental evidence supports this hypothesis, with transgenic mice expressing the CADASIL R90C mutation, specifically in VSMCs, recreating the classic CADASIL pathology. These findings underline the essential role of Notch3 in the function and integrity of VSMCs [136]. Notch3 is also expressed in pericytes. However, its role in pericyte function remains controversial. Henshall et al. reported that Notch3^−/−^ knockout mice showed BBB disruption and leakage but found no impact on pericytes [136]. In contrast, Wang et al. demonstrated that Notch3 is necessary for pericyte proliferation in the zebrafish brain, indicating a potential role in pericyte dynamics [137]. These conflicting findings highlight the need for further research to clarify the role of Notch3 in pericyte biology. A significant connection between Notch3 and PDGFR-β has been established. Notch3 regulates PDGFR-β expression, and Notch3^−/−^ mice exhibit reduced levels of this receptor and AVMs [138,139]. Similarly, PDGFR-β levels are decreased in individuals with CADASIL [138]. Since PDGFR-β is expressed by both ECs and mural cells, including pericytes and VSMCs, its role in vascular health is critical. Studies have shown that *Pdgf-b*^−/−^ mice lack pericytes in their vasculature, leading to microaneurysm formation and rupture during gestation [140]. Heterozygous *Pdgfr*-β mice exhibit reduced pericyte numbers, suggesting a direct correlation between Pdgfr-β expression and pericyte density [141]. This evidence further supports the intricate relationship between Notch3 signaling, PDGFR-β levels, and mural cell function. While the role of Notch3 in VSMC maintenance and vascular integrity is well established, its effect on pericytes remains less clear, with contradictory findings warranting further investigation.

Moyamoya disease (MMD), a progressive vasculopathy characterized by the narrowing and eventual occlusion of the intracranial internal carotid arteries, is a significant cause of childhood stroke. Rare, damaging variants in two VSMCs-specific genes have been found in non-East Asian patients with MMD: *DIAPH1* (diaphanous-1), a critical effector of actin remodeling in vascular cells [142], and *ACTA2*, which encodes the smooth muscle-specific isoform of a-actin in VSMCs [143]. Both genes converge on pathways regulating the cytoskeleton, highlighting their essential role in maintaining vascular integrity and function.

## 10. Future Perspectives

The intricate interplay between components of the NVU and GM underscores their central role in neurogenesis, vascular development, and brain maturation during critical periods of fetal and neonatal growth. This complexity also highlights their vulnerability to perinatal ischemic injuries. Understanding the dynamic and maturation-dependent pathophysiological responses within the NVU is essential to developing targeted therapeutic strategies. By focusing on age-specific mechanisms—such as the modulation of the BBB, neurovascular coupling, and astrocyte–pericyte–endothelial interactions—future research and clinical interventions can better preserve neural integrity and promote repair in the developing brain.

Stem cell-based therapies have shown promising results at various levels, but most remain in the preclinical stage or have been tested on a limited number of patients. A significant gap exists in pediatric studies, which may be addressed while considering the challenges of researching such a vulnerable population. Despite these limitations, stem cell therapies hold promise for mitigating the long-term impact of perinatal stroke and fostering resilience in the CNS.

Genetic insights into NVU-associated disorders offer valuable opportunities to explore novel therapeutic strategies. For instance, rare damaging variants in *DIAPH1* and *ACTA2* in non-East Asian MMD patients emphasize the critical role of actin cytoskeleton regulation in vascular integrity and function. Understanding these pathways could guide the development of targeted interventions to prevent or manage cerebrovascular diseases. Similarly, targeting specific junction proteins (e.g., ESAM, JAMs, occludin) or ECM degradation pathways mediated by MMPs may mitigate BBB disruption in ischemic and hemorrhagic injuries. Moreover, therapies focusing on Notch3 signaling, PDGFR-β, and the modulation of mural cell maintenance have potential applications for conditions like CADASIL and MMD.

Emerging therapeutic modalities, including lipid peroxidation inhibitors (e.g., ferrostatin-1), immunomodulators, and gene-editing technologies, offer additional avenues for addressing both acquired and genetic NVU dysfunctions. Preclinical models remain critical to evaluating these interventions, and approaches utilizing humanized models- such as patient-derived 3D NVU models- may significantly enhance the testing of novel therapies. However, translating these findings into clinical practice will require robust multicenter trials focused on safety, efficacy, and long-term outcomes.

## Figures and Tables

**Figure 1 ijms-26-01886-f001:**
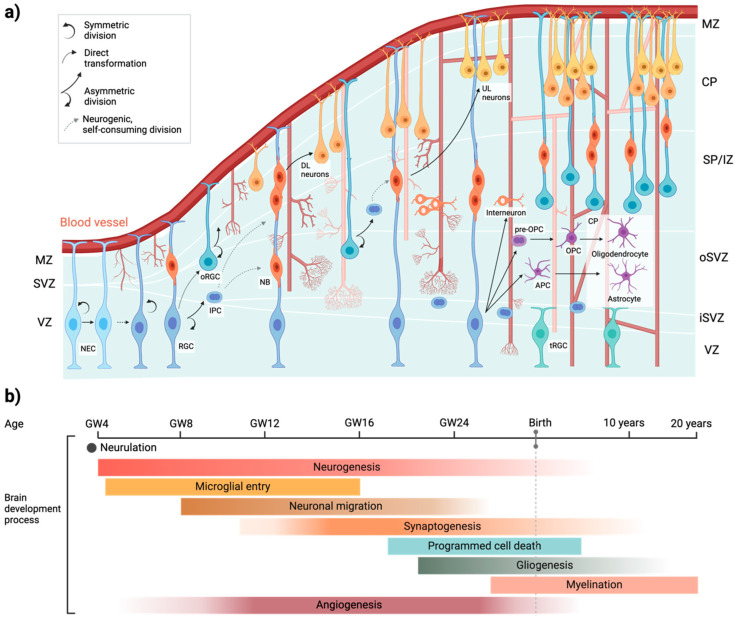
**Key stages of human brain development: from conception to adulthood.** (**a**) Neurovascular communication during CNS development. Neocortical development begins with neurogenesis, where neural progenitor cells differentiate into neuroblasts and eventually into neurons, oligodendrocytes, and astrocytes. Simultaneously, the formation of the cerebral vasculature is initiated as sprouting blood vessels from the perineural vascular plexus extend radial branches toward the ventricular zone. These initial branches further subdivide (anastomosis) and interconnect, forming a complex and intricately organized vascular network essential for supporting the metabolic demands of the developing neocortex. APC, astrocyte precursor cell; CP, cortical plate; DL, deep layer; IPC, intermediate progenitor cell; iSVZ, inner SVZ; IZ, intermediate zone; MZ, marginal zone; NB, neuroblast; NEC, neuroepithelial cell; OPC, oligodendrocyte precursor cell; oRGC, outer radial glial cell; oSVZ, outer SVZ; RGC, radial glial cell; SP, sublplate; SVZ, subventricular zone; tRGC, truncated radial glial cell; UL, upper layer; VZ, ventricular zone. (**b**) This diagram highlights the major stages and timeline of human brain development, starting with neurulation, followed by neurogenesis, microglial entry, neuronal migration, and synaptogenesis. By 18 GW, apoptosis eliminates excess cells to refine populations and establish proper synaptic connectivity. During the third trimester, glial cells support the myelination of neurons, a process that continues postnatally into adulthood. Angiogenesis, essential for brain development, begins in the early embryonic stages, peaks during rapid growth, stabilizes by birth, and continues postnatally with limited adaptations, supporting brain function throughout life.

**Figure 2 ijms-26-01886-f002:**
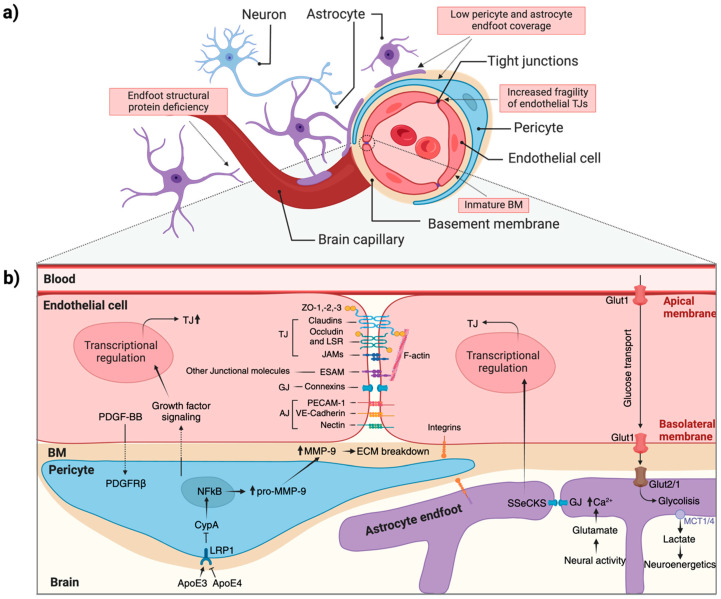
**Graphical representation of the BBB structure.** (**a**) Schematic representations of a cross-section of a CNS capillary, with key characteristics observed during the perinatal period highlighted in red. (**b**) NVU unit cell–cell interaction. ECs are connected with each other through the TJ, AJ, and GJ proteins, together with cytoplasmatic proteins, ZO maintaining BBB function. Pericytes partially cover the ECs and are embedded in the basement membrane. The pericytes act as contractile components in the brain’s microvascular BBB, sensible to astrocyte inputs. AJ, adherent junctions; BM, basement membrane; ECM, extracellular matrix; GJ, gap junctions; TJ, tight junctions; ZO, zonula occludens. Blunt arrows (┴) indicate inhibition while sharp arrows (→) indicate stimulation.

**Figure 3 ijms-26-01886-f003:**
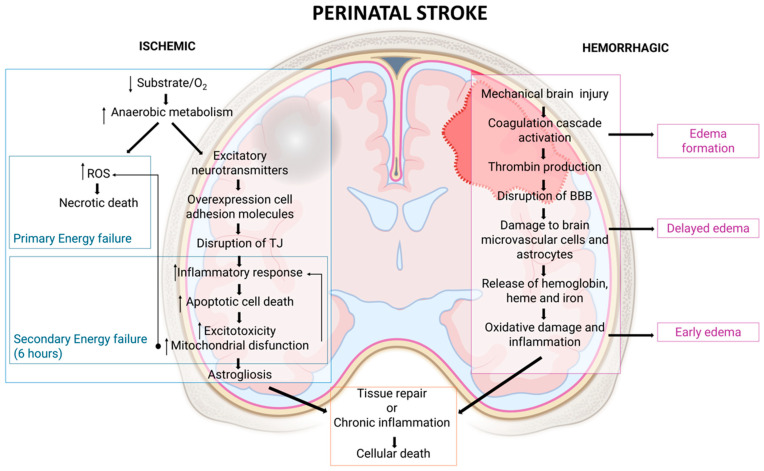
**Schematic overview of the mechanisms underlying brain damage after a perinatal stroke.** Hypoxic–ischemic injury (**left**) involves energy failure, ROS production, inflammation, and cell death. Intracerebral hemorrhagic injury (**right**) includes BBB disruption, thrombin activation, oxidative stress, and edema formation. Both lead to tissue repair or chronic inflammation and cell death. ROS, reactive oxygen species; TJ, tight junctions; BBB, blood–brain barrier.

**Figure 4 ijms-26-01886-f004:**
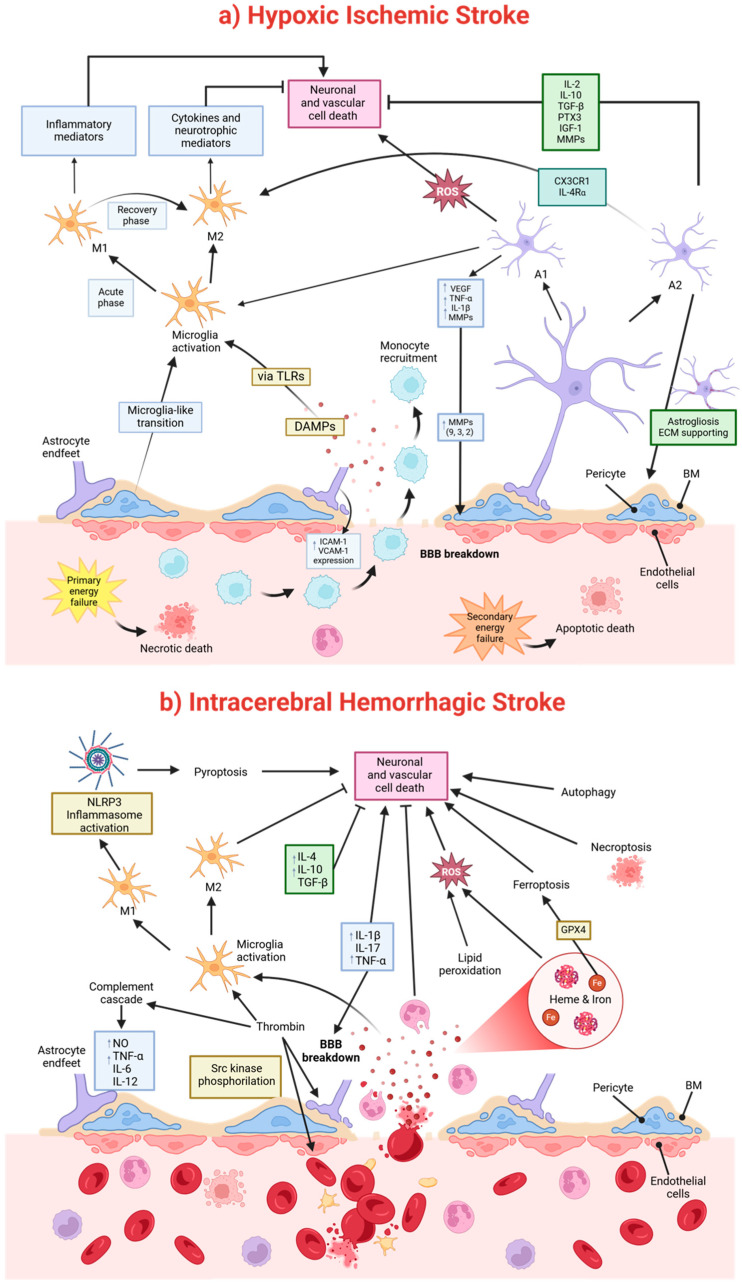
**Key mechanisms underlying hypoxic–ischemic (HI) stroke and neonatal intracerebral hemorrhagic stroke (ICH).** (**a**) HI stroke is characterized by energy failure, leading to both necrotic and apoptotic cell death. Pro-inflammatory astrocytes (A1) and microglia (M1) exacerbate damage through cytokine release, while neuroprotective astrocytes (A2) and microglia (M2) facilitate recovery by providing extracellular matrix (ECM) support and promoting anti-inflammatory signaling. (**b**) ICH is defined by thrombin activation, oxidative stress caused by heme and iron deposition, and activation of cell death pathways such as pyroptosis, necroptosis, and ferroptosis, which collectively contribute to neuronal and vascular damage. BM, basement membrane; DAMPs, Damage-Associated Molecular Patterns; ECM, extracellular matrix; Glutathione Peroxidase 4; ICAM-1, Intercellular Adhesion Molecule 1; M1, pro-inflammatory microglia; M2, anti-inflammatory microglia; NLRP3, Pyrin Domain-Containing Protein 3; ROS, Reactive Oxygen Species, GPX4, TLRs, Toll-Like Receptors; VCAM-1, Vascular Cell Adhesion Molecule 1; VEGF, Vascular Endothelial Growth Factor. Blunt arrows (┴) indicate inhibition while sharp arrows (→) indicate induction.

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
