# Peer review of "The Triad of Blood–Brain Barrier Integrity: Endothelial Cells, Astrocytes, and Pericytes in Perinatal Stroke Pathophysiology"

_ijms, 2025, doi:10.3390/ijms26051886_

Round 1

Reviewer 1 Report

Comments and Suggestions for Authors

First class review. see attached

Author Response

Dear Reviewer,

We sincerely appreciate your thoughtful and positive feedback on our review article. We are delighted to hear that you found our discussion on the neurovascular unit (NVU) in perinatal stroke to be well-organized, logically structured, and engaging. Your comments on the clarity of our writing and the quality of our figures are highly encouraging.

Your endorsement of our work further motivates us to continue contributing to this field. 

Thank you once again for your time and valuable insights.

Sincerely yours,

Cristòfol Vives-Bauza, PhD.

Reviewer 2 Report

Comments and Suggestions for Authors

Dear Authors,

Your manuscript presents a comprehensive review of the role of the NVU in perinatal stroke, highlighting its pathophysiological mechanisms and therapeutic opportunities. However, I would suggest the following:

1.     Methodology Section: Provide a clear description of how literature was selected and analyzed (prisma?)

2.     Discussion& Clarity: I would suggest to structure the discussion with subheadings and integrate clinical relevance more explicitly.

3.     Conclusion: Iwould suggest to strengthen the summary of key findings and implications for future research.

Thank you

Best regards

Author Response

Thank you very much for your review of the Manuscript entitled “The Triad of Blood Brain Barrier Integrity: Endothelial Cells, Astrocytes, and Pericytes in Perinatal Stroke Pathophysiology”. We appreciate your comments and we have modified our Manuscript according to your concerns as follows:

Comments 1. Methodology Section: Provide a clear description of how literature was selected and analyzed (prisma?)

Response 1: Thank you for your valuable feedback. In this review, we conducted a targeted literature search focusing on key concepts related to the neurovascular unit (NVU) and the blood-brain barrier (BBB). We identified relevant articles through a systematic search in PubMed using Boolean operators, with a focus on publications from the past five years. Our search terms included "Pediatric stroke," "Perinatal stroke," "Brain development," "Angiogenesis," "Neurogenesis," "Germinal matrix," "Blood-brain barrier," "Astrocyte-endothelial interactions," "Intracerebral hemorrhagic stroke," "Ischemic stroke," and "Stroke genetics."

To ensure a comprehensive perspective, we also reviewed reference lists of selected articles to include both foundational studies and recent advancements. While we aimed for a thorough and structured literature review, we emphasize that this manuscript is a narrative review rather than a systemic review or meta-analysis. Therefore, we did not follow PRISMA guidelines but focused on synthesizing the most relevant and high-impact literature.

 approach enabled us to compile an up-to-date and well-supported discussion based on high-impact scientific literature.

We appreciate your suggestion and are open to incorporating any modifications the reviewers deem necessary to further enhance the clarity and rigor of our manuscript.

Comments 2: Discussion & Clarity: I would suggest to structure the discussion with subheadings and integrate clinical relevance more explicitly.

Response 2: We sincerely appreciate the reviewer’s suggestion to enhance the structure and clinical relevance of our dscusion. To strengthen the clinical applicability of our review, we have incorporated additional details on the management and diagnosis of pediatric stroke in the introduction (section: Pediatric Stroke, Page 3, lines 73-96): “Pediatric strokes, like those in adults, are classified by cause as either ischemic or hemorrhagic [11]. Ischemic strokes include arterial ischemic stroke (AIS) and venous infarction, typically caused by cerebral or cortical vein thrombosis (CSVT) [8,9]. These strokes are further categorized by timing: fetal ischemic stroke (diagnosed before birth), neonatal ischemic stroke (within 28 days of birth), and presumed perinatal ischemic stroke (PIS, diagnosed later but presumed to have occurred between the 20th week of gestation and 28 days postnatal) [12,13]. Hemorrhagic strokes, on the other hand, can manifest as intracerebral (ICH), intraventricular (IVH), or subarachnoid (SAH) hemorrhage. They are classified as either primary (resulting from vascular anomalies or bleeding disorders) or secondary (arising from ischemic infarction) [8,14]. Distinguishing these stroke types may be challenging without prompt imaging, particularly in neonates, where acute presentations may be missed [12,15,16]. Recognizing stroke at different ages is crucial for timely intervention, as early treatment can help preserve brain function and support recovery. The impact of any stroke is influenced by the neurodevelopmental stage at the time of the event [7], underscoring the importance of early detection and management by pediatric health professionals.”

Additionally, we have added a new subsection in the discussion, titled “Emerging cell-based therapeutic strategies for pediatric stroke”, where we review recent data on the potential of stem-cell based therapies in pediatric stroke treatment. This new section can be found in the revised manuscript on Page 10, lines 338-363:

Emerging cell-based therapeutic strategies for pediatric stroke.

Strategies aimed at promoting the anti-inflammatory polarization of both microglia and astrocytes or inhibit harmful pro-inflammatory responses hold promise for the management of both ischemic and hemorrhagic pediatric stroke. In line with these strategies, cell-based treatments using mesenchymal stem cells (MSCs), or umbilical cord blood-derived mesenchymal stem cell (UC-MSCs) transplantation show potential, as they possess immunosuppressive and anti-inflammatory properties that may have protective and regenerative effects (reviewed in [104,105]). MSCs modulate the immune response by regulating the function of immune cells, such as T-cells and B-cells, macrophages and dendritic cells [106]. In neonatal rodents, intraventricular transplantation of UC-MSCs has been shown to reduce post-hemorrhagic hydrocephalus and brain injury following IVH [107]. In addition to their anti-inflammatory properties, studies in animal models have demonstrated that both MSCs and UC-MSCs promote neuroplasticity by activating NMDA and AMPA receptors. This activation, in turn, stimulates the expression of various trophic factors, such as VEGF [108,109]. Through this mechanism, MSCs and UC-MSCs may also contribute to angiogenesis, EC proliferation and astrocytes differentiation -processes known to be regulated by VEGF. Furthermore, MSCs also exert neuroprotective effects through mitochondrial transfer and the regulation of ROS production [110].

Other cells with neuroprotective potential include endothelial progenitor cells (EPCs), which secrete protective cytokines and growth factors. These factors support the self-repair of injured ECs and help restore their structure and function by integrating into damaged areas [111]. Another promising cell source for therapy is induced pluripotent stem cell (iPSC)-derived progenitors. The administration of iPSC-derived neural progenitor and precursor cells has been shown to enhance sensorimotor function, reduce lesion volume, and promote neurogenesis and angiogenesis [112]. Additionally, positive effects have been observed with iPSC-derived MSCs, inducing angiogenesis [113], as well as with neuroepithelial-like stem cells, which promote neuronal regeneration [114]. Overall, regenerative cells may contribute to a favorable environment for tissue repair, ultimately improving functional outcomes after perinatal stroke.”

Comments 3: Conclusion: I would suggest strengthening the summary of key findings and implications for future research.

Response 3: Following reviewer’s suggestion, we have added more information in the Future Perspective section, Page 14, lines 485-489: “Stem cell-based therapies have shown promising results at various levels, but most remain in the preclinical stage or have been tested on a limited number of patients. A significant gap exists in pediatric studies, which may be addressed while considering the challenges of researching such a vulnerable population. Despite these limitations, stem cell therapies hold promise for mitigating the long-term impact of perinatal stroke and fostering resilience in the CNS”.

And we have also modified in the same section the final paragraph, Page 15, lines504-510: “Emerging therapeutic modalities, including lipid peroxidation inhibitors (e.g., ferrostatin-1), immunomodulators, and gene-editing technologies, offer additional avenues for addressing both acquired and genetic NVU dysfunctions. Preclinical models remain critical to evaluating these interventions, and approaches utilizing humanized models- such as patient-derived 3D NVU models- may significantly enhance the testing of novel therapies. However, translating these findings into clinical practice will require robust multicenter trials focused on safety, efficacy, and long-term outcomes.”

We believe these additions improve the clarity and clinical impact of our review, and we appreciate the reviewer’s valuable input.

Reviewer 3 Report

Comments and Suggestions for Authors

Please find my comments below:

  • Add a structured summary table highlighting major NVU dysfunction mechanisms and related consequences in perinatal stroke.

  • Expand the discussion on potential therapeutic interventions, emphasizing experimental models and clinical trial data.

  • Provide additional insights into the long-term effects of perinatal stroke on neurodevelopment and brain plasticity.

  • Ensure concise descriptions of molecular pathways while maintaining necessary details.

  • Review terminology to ensure consistency and clarity across sections.

Author Response

Thank you very much for your review of the Manuscript entitled “The Triad of Blood Brain Barrier Integrity: Endothelial Cells, Astrocytes, and Pericytes in Perinatal Stroke Pathophysiology”. We appreciate your comments and we have modified our Manuscript according to your concerns as follows:

Comments 1. Add a structured summary table highlighting major NVU dysfunction mechanisms and related consequences in perinatal stroke.

Response 1: We appreciate the reviewer’s suggestion. However, we believe that a structured summary table would be redundant given the information already presented in Figure 3. This figure graphically summarizes the cellular and molecular events following ischemic and hemorrhagic perinatal stroke, their impact on NVU structure and function, and the resulting clinical outcomes. We believe this visual representation effectively conveys the key mechanisms and consequences of NVU dysfunction in perinatal stroke.

Comments 2: Expand the discussion on potential therapeutic interventions, emphasizing experimental models and clinical trial data.

Response 2: We sincerely appreciate the reviewer’s suggestion to enhance the clinical relevance of our discussion. To strengthen the clinical applicability of our review, we have incorporated a new subsection in the discussion, titled “Emerging cell-based therapeutic strategies for pediatric stroke”, where we review recent data on the potential of stem-cell based therapies in pediatric stroke treatment. This new section can be found in the revised manuscript on Page 10, lines 338-363:

Emerging cell-based therapeutic strategies for pediatric stroke.

Strategies aimed at promoting the anti-inflammatory polarization of both microglia and astrocytes or inhibit harmful pro-inflammatory responses hold promise for the management of both ischemic and hemorrhagic pediatric stroke. In line with these strategies, cell-based treatments using mesenchymal stem cells (MSCs), or umbilical cord blood-derived mesenchymal stem cell (UC-MSCs) transplantation show potential, as they possess immunosuppressive and anti-inflammatory properties that may have protective and regenerative effects (reviewed in [104,105]). MSCs modulate the immune response by regulating the function of immune cells, such as T-cells and B-cells, macrophages and dendritic cells [106]. In neonatal rodents, intraventricular transplantation of UC-MSCs has been shown to reduce post-hemorrhagic hydrocephalus and brain injury following IVH [107]. In addition to their anti-inflammatory properties, studies in animal models have demonstrated that both MSCs and UC-MSCs promote neuroplasticity by activating NMDA and AMPA receptors. This activation, in turn, stimulates the expression of various trophic factors, such as VEGF [108,109]. Through this mechanism, MSCs and UC-MSCs may also contribute to angiogenesis, EC proliferation and astrocytes differentiation -processes known to be regulated by VEGF. Furthermore, MSCs also exert neuroprotective effects through mitochondrial transfer and the regulation of ROS production [110].

Other cells with neuroprotective potential include endothelial progenitor cells (EPCs), which secrete protective cytokines and growth factors. These factors support the self-repair of injured ECs and help restore their structure and function by integrating into damaged areas [111]. Another promising cell source for therapy is induced pluripotent stem cell (iPSC)-derived progenitors. The administration of iPSC-derived neural progenitor and precursor cells has been shown to enhance sensorimotor function, reduce lesion volume, and promote neurogenesis and angiogenesis [112]. Additionally, positive effects have been observed with iPSC-derived MSCs, inducing angiogenesis [113], as well as with neuroepithelial-like stem cells, which promote neuronal regeneration [114]. Overall, regenerative cells may contribute to a favorable environment for tissue repair, ultimately improving functional outcomes after perinatal stroke.”

Accordingly, we have added more information in the Future Perspective section, Page 14, lines 485-489: “Stem cell-based therapies have shown promising results at various levels, but most remain in the preclinical stage or have been tested on a limited number of patients. A significant gap exists in pediatric studies, which may be addressed while considering the challenges of researching such a vulnerable population. Despite these limitations, stem cell therapies hold promise for mitigating the long-term impact of perinatal stroke and fostering resilience in the CNS”.

And we have also modified in the same section the final paragraph, Page 15, lines 504-510: “Emerging therapeutic modalities, including lipid peroxidation inhibitors (e.g., ferrostatin-1), immunomodulators, and gene-editing technologies, offer additional avenues for addressing both acquired and genetic NVU dysfunctions. Preclinical models remain critical to evaluating these interventions, and approaches utilizing humanized models- such as patient-derived 3D NVU models- may significantly enhance the testing of novel therapies. However, translating these findings into clinical practice will require robust multicenter trials focused on safety, efficacy, and long-term outcomes.”

We believe these additions improve the clarity and clinical impact of our review, and we appreciate the reviewer’s valuable input.

Comments 3: Provide additional insights into the long-term effects of perinatal stroke on neurodevelopment and brain plasticity.

Response 3: To address this point, we have expanded the Human Brain Development section by adding a sentence that contextualizes the role of brain plasticity in perinatal stroke recovery and long-term outcomes. Specifically, on Page 4, lines 141-144, we now state: “Postnatally, the brain’s structure and plasticity continue to evolve, enabling lifelong adaptation and learning [2,5]. Because plasticity in the developing brain is highly dynamic and reaches its peak within the first two years of life, children generally exhibit a greater capacity for stroke recovery compared to adults [35].”

Comments 4: Ensure concise descriptions of molecular pathways while maintaining necessary details. Review terminology to ensure consistency and clarity across sections.

Response 4: In line with the reviewer’s suggestion, we have carefully reviewed the entire manuscript and made revisions to ensure concise descriptions of molecular pathways while maintaining necessary details. Additionally, we have refined terminology for consistency and improved overall clarity and flow.